# Type III Secretion System-Mediated Induction of Systemic Resistance by *Pseudomonas marginalis* ORh26 Enhances Sugar Beet Defence Against *Pseudomonas syringae* pv. *aptata*

**DOI:** 10.3390/plants14111621

**Published:** 2025-05-26

**Authors:** Marija Nedeljković, Aleksandra Mesaroš, Marija Radosavljević, Nikola Đorđević, Slaviša Stanković, Jelena Lozo, Iva Atanasković

**Affiliations:** 1University of Belgrade—Faculty of Biology, 11000 Belgrade, Serbia; marija.nedeljkovic@bio.bg.ac.rs (M.N.); aleksandra.mesaros@bio.bg.ac.rs (A.M.); m41_2022@stud.bio.bg.ac.rs (M.R.); slavisas@bio.bg.ac.rs (S.S.); jlozo@bio.bg.ac.rs (J.L.); 2Centre for Biological Control and Plant Growth Promotion, University of Belgrade—Faculty of Biology, 11000 Belgrade, Serbia; 3Agrounik Ltd., 11000 Belgrade, Serbia; nikola.djordjevic@agrounik.rs

**Keywords:** type III secretion system, *Pseudomonas*, induced systemic resistance, sugar beet, plant–microbe interactions, biotic stress

## Abstract

The increasing demand for sustainable agricultural practises has sparked interest in microbes that promote plant immunity. Among these, *Pseudomonas* species have shown the potential to enhance induced systemic resistance (ISR) in crops. While type III secretion systems (T3SSs) in pathogenic bacteria have been widely studied for their role in local immunosuppression, their function in beneficial *Pseudomonas* species and on a systemic level remains largely unexplored. We show for the first time that the T3SS of a plant-beneficial *Pseudomonas* strain induces ISR by root colonisation. T3SS-positive *Pseudomonas* isolates were applied to the roots of sugar beet (*Beta vulgaris* L.) and systemic effects on plant immunity were assessed in leaves exposed to the pathogen *P. syringae* pv. *aptata* P21. Our results show that *P. marginalis* ORh26 reduced lesion size and pathogen proliferation in sugar beet leaves. ORh26 activated peroxidase and phenylalanine ammonia-lyase and upregulated *NPR1* and *MYC2* defence genes. Remarkably, a T3SS-deficient mutant of ORh26 failed to induce these effects. Genomic analysis identified T3SS structural genes and effector proteins, including a pectate lyase and an effector of the HopJ family, that may mediate these responses. This study reveals a previously uncharacterised role of T3SS in the induction of ISR and improves our understanding of plant–microbe interactions.

## 1. Introduction

As agriculture worldwide shifts to sustainable practises, it is becoming increasingly important to understand the mechanisms by which beneficial bacteria boost plant immunity. Harnessing the molecular machinery of these microbes offers a promising alternative to synthetic agrochemicals. Over-reliance on agrochemicals has led to pollution, soil degradation and a decline in biodiversity, necessitating a switch to environmentally friendly alternatives. One promising solution is the use of beneficial microorganisms to boost the immunity of plants through induced systemic resistance (ISR). ISR is a mechanism by which some plant-beneficial bacteria and fungi in the rhizosphere prime the whole plant body for enhanced defence against different pathogens and insect herbivores [1]. In contrast to chemical treatments, ISR prepares plants for defence against pathogens by activating their innate immune system without harming the environment. Beneficial microbes inhabiting plant roots can stimulate resistance through the release of microbial-associated molecular patterns (MAMPs) and other signalling molecules that trigger systemic plant defence mechanisms [2]. Plant protection from biotic stress by ISR not only reduces dependence on synthetic agrochemicals, but is also in line with global efforts to promote sustainable agriculture while maintaining plant productivity. The genus *Pseudomonas* comprises a variety of beneficial plant-associated bacteria that induce a strong ISR response and represent a sustainable alternative for plant protection [3]. These bacteria can trigger plant immunity through the release of MAMPs such as flagella, lipopolysaccharides and siderophores that are recognised by plant pattern recognition receptors (PRRs) such as Flagelin Sensitive 2 (FLS2), which bind to conserved peptides such as flg22 and activate a defence signalling cascade [4]. Many ISR-inducing factors are unidentified, highlighting the need for further research to uncover the full repertoire of ISR determinants in *Pseudomonas* and their interactions with plant defence systems.

The type III secretion system (T3SS) is a syringe-like nanomachine used by *Pseudomonas* and other Gram-negative bacteria to inject effector proteins (T3Es) directly into host cells. It consists of the cytoplasmic complex that drives and regulates secretion, the export apparatus that selects and translocates the effectors, the basal body that spans the inner and outer bacterial membranes and anchors the structure, and the needle filament that extends outward to form a translocation pore in the host cell membrane through which the effectors pass directly into the host cytoplasm [5]. Interestingly, T3SS has a common evolutionary origin with the flagella [6], whose components are known ISR inducers. However, T3SS has never been explored as a source of MAMPs and the potential role of T3SS in triggering ISR has not yet been explored. T3SS is found in both pathogenic and non-pathogenic members of the genus *Pseudomonas*. In pathogens such as *P. syringae*, the role of T3SS as a virulence factor is well documented. Its effector proteins can locally suppress host immune responses and thus facilitate infection [5]. While T3SS is well characterised in pathogenic *Pseudomonas* due to its role in local immunomodulation, its function in positive interactions between plants and microbes and its effects on a systemic level remain unclear. Recent studies suggest a possible immunomodulatory role [7], but there is no evidence that T3SS activity in beneficial bacteria is associated with systemic resistance mechanisms. We therefore hypothesise that T3SS in beneficial *Pseudomonas* isolates can contribute to ISR either by serving as a source of MAMPs or by modulating plant immune and hormonal signalling pathways through T3Es. Understanding this potential role of T3SS in non-pathogens could reveal new mechanisms by which beneficial bacteria enhance plant defence mechanisms.

In a previous study, a collection of T3SS-positive *Pseudomonas* isolates from the sugar beet (*Beta vulgaris* L.) microbiome was established and the functionality of the T3SS in these isolates was confirmed [8]. In particular, the expression of T3SS was shown to be active and responsive to sugar beet-derived signals. Furthermore, deletion of the T3SS in one isolate, *P. marginalis* OL141, led to phenotypic effects on the host plant: the aerial parts of sugar beet showed reduced growth when the roots were treated with the T3SS-deficient mutant compared to the wild type, indicating a systemic effect. Building on these findings, the current aim is to further characterise the systemic effect of the isolate collection by assessing its impact on ISR against a pathogen that uses the T3SS as a virulence factor—*P. syringae*. More specifically, *P. syringae* pv. *aptata* P21 is used as a model pathogen, since it is a significant agricultural treat to sugar beet [9]. For *P. marginalis* ORh26, an isolate identified as strong ISR inducer, the role of T3SS in ISR induction is investigated using a T3SS-deficient mutant and by analysing the activity of ISR-associated molecular components. This research is crucial to our understanding of the molecular mechanisms underlying plant defence mechanisms and opens important questions about the mechanism of T3SS-dependent ISR induction, potentially paving the way for the development of innovative, microbe-based strategies for sustainable crop protection.

## 2. Results

### 2.1. Effects of T3SS-Positive Pseudomonas Isolates on Sugar Beet Resistance to Pseudomonas syringae pv. aptata P21

Non-pathogenic *Pseudomonas* species living on plants are known to induce systemic resistance and help plants to defend themselves against pathogens [2]. To determine whether T3SS-positive *Pseudomonas* isolates can induce ISR in sugar beet, we tested a collection of T3SS-positive strains (Table 1) previously isolated from healthy sugar beet plants by Krstić Tomić et al. [10].

The presence and expression of T3SS in these strains was confirmed by Nedeljković et al. [8]. As a pathogen, we used *P. syringae* pv. *aptata* P21, which utilises T3SS as a virulence factor and is a significant agricultural threat, causing substantial damage to sugar beet crops and leading to considerable economic losses [9]. To evaluate the effects of the isolates, sugar beet plants were treated by adding individual *Pseudomonas* isolates to the soil so that the bacteria could interact with the roots. Untreated plants served as a control group. After treatment, leaves were infected with *P. syringae* pv. *aptata* P21 and susceptibility to infection was assessed by the size of necrotic lesions at the infection sites (Figure 1A). Lesion area was used as a phenotypic proxy for disease severity during the initial screening of candidate ISR-inducing strains. This parameter was used in the initial screen as it reflects both symptom severity and the plant’s immune response, making it a suitable indicator of ISR effectiveness [11,12,13]. Treatment with *P. marginalis* Orh26 significantly reduced lesion size compared to the control group, indicating the induction of systemic resistance. None of the other isolates tested had a significant effect on lesion size. Figure 1B shows representative images of the necrotic lesions in the control group and the Orh26-treated group.

The lesions in the Orh26-treated plants appeared as small, localised spots, indicating the limited spread of the pathogen, while larger, more compact lesions were observed in the untreated control group. These results show that *P. marginalis* Orh26 is the only T3SS-positive isolate tested that has a significant effect on systemic resistance in sugar beet. This made it the focus for further research into T3SS functionality in ISR.

### 2.2. Differential Immune Responses Across Treatment Conditions

In addition to lesion size, we also analysed the hydrogen peroxide (H_2_O_2_) content and protective enzyme activity in both uninfected and *P. syringae*-infected plants to assess the effects of the different *Pseudomonas* isolates on the immune response of the plants. Figure 2 shows the ratio between infected and uninfected plants for each of the measured parameters, allowing us to assess whether the parameters were induced or suppressed by the infection. Appendix A shows the data for each condition separately. These parameters are important indicators of plant resistance mechanisms, as changes in their values can signal the activation of immune pathways. H_2_O_2_ (Figure 2A) is an important signalling molecule in plant defence, which is often produced as a reactive oxygen species (ROS) in response to pathogen attack and can trigger the activation of downstream immune responses. Peroxidase (POD; Figure 2B) is an enzyme involved in the regulation of H_2_O_2_ and ROS levels and is often associated with the strengthening of cell walls during defence responses. Phenylalanine ammonia-lyase (PAL; Figure 2C) is an enzyme of phenylpropanoid metabolism whose activity is associated with the biosynthesis of phytoalexins and other compounds that contribute to resistance to pathogens. Polyphenol oxidase (PPO; Figure 2D) is involved in the oxidation of phenolic compounds that can help strengthen plant cell walls and protect against pathogens. In the control group that was not treated with bacteria, all four parameters were induced upon infection with *P. syringae* pv. *aptata* P21, indicating an active immune response.

However, the plants treated with the different non-pathogenic *Pseudomonas* isolates showed different effects. For example, in plants treated with *P. kilonensis* Orh244, all tested parameters were inhibited after infection, suggesting that this isolate may suppress the plant’s immune response. In contrast, *P. marginalis* Orh26 showed a different pattern: it significantly increased POD activity compared to the control group, which was accompanied by lower H_2_O_2_ levels, indicating a possible balance between oxidative stress and defence activation. In addition, Orh26 induced higher PAL activity than the control group. Looking at the individual parameters, H_2_O_2_ was induced in most treatments, except in the Orh244-treated plants. POD was significantly stimulated only by Orh26, while other isolates did not induce this enzyme compared to the control. PAL was induced in most treatments, with Orh26 showing the strongest stimulatory effect. Finally, PPO was not induced in most treatments, but *P. lurida* ML159 stood out by inducing PPO activity compared to the control, although it did not significantly affect the other enzymes. These results suggest that different *Pseudomonas* isolates can modulate components of the plant immune response differently and that *P. marginalis* Orh26, which induces both POD and PAL activity, may be particularly effective in enhancing sugar beet resistance to *P. syringae* due to the induction of these particular components.

### 2.3. Characterisation of T3SS Genes in the Pseudomonas marginalis Orh26 Genome

Strain Orh26 was initially classified as *P. grimontii* based on the partial sequence of its 16S rRNA gene [8]. Whole-genome sequencing and comparative analysis of several housekeeping genes (Appendix A) led to the reclassification of the strain as *P. marginalis* ORh26. Phylogenetic analysis of housekeeping genes revealed that ORh26 is most similar to *P. marginalis* MGMM3 (NCBI Reference Sequence: NZ_CP123957.1). To further confirm its taxonomic placement, we conducted a whole-genome phylogenomic analysis using the Type Strain Genome Server (TYGS) platform [14]. The closest match was *P. marginalis* DSM 13124, with digital DNA–DNA hybridisation (dDDH) values of 74.9% (d0), 69.5% (d4) and 76.5% (d6), and a G+C content difference of only 0.29%. These values place ORh26 within the species boundary of *P. marginalis*, validating its reclassification. The TYGS tree also confirmed clustering with *P. marginalis* reference strains (Appendix A). The genome of *P. marginalis* ORh26 was assembled into 29 scaffolds, totalling approximately 6.7 Mb in size, with a GC content of 60.78%, and 6134 predicted protein-coding genes. These features are broadly consistent with other *P. marginalis* strains, such as DSM 13124 and ICMP 3553, which also encode type III secretion systems.

A single complete T3SS operon belonging to the Hrc1 family was identified in the ORh26 genome. The structural genes within the T3SS operon were mapped and compared to those of *P. marginalis* SBW25, a beneficial bacterium with a functional T3SS [15], and *P. syringae* pv. tomato DC3000 (NCBI Reference Sequence: NC_004578.1), a known plant pathogen [16]. In general, the T3SS genes of ORh26 show a sequence identity of more than 70% with the corresponding genes in SBW25 (Appendix A). Additional secretion systems (T1SS–T6SS) were also detected (Appendix A).

The structural genes of the T3SS operon in ORh26 span a 15.8 kb region and comprise 20 genes. It was found that the T3SS operon in ORh26 is larger than in SBW25 and contains two additional genes, *hrcD* and *hrcV* (Figure 3). However, it is smaller than the operon in DC3000 because several structural genes are missing in ORh26, including *hrpK1*, *hrpF*, *hrpZ1* and *hrpA1*. These missing genes encode components of the T3SS pilus, such as subunits that form the filament structure and proteins that comprise the translocon in the host cell membrane. The only needle component annotated in ORh26 is *hrpB*, which encodes the inner rod at the base of the needle filament. In addition, several genes encoding hypothetical proteins could be annotated at this end of the operon.

A total of 26 genes were predicted as putative T3Es in the ORh26 genome (Appendix A). Among these, three proteins have a high probability of being T3Es (Table 2). Based on domain organisation, one was identified as a pectate lyase, another as homologous to HopJ effectors in other *Pseudomonas* species and a third as a hypothetical protein with putative DNA-binding domains.

### 2.4. Induction of ISR and Peroxidase Activity Are T3SS-Dependent

The aim of this experiment was to investigate whether the T3SS in *P. marginalis* ORh26 plays a role in the observed activation of systemic immunity in sugar beet, as it is not known whether T3SS can act as a ISR regulator. To address this question, we generated a mutant ORh26 strain lacking the T3SS by deleting the *hrcT* gene. The *hrcT* gene encodes a component of the T3SS export apparatus, and its deletion leads to a complete loss of T3SS functionality [17]. We then compared the immune responses in sugar beet plants treated with the wild-type strain (wt) and the T3SS mutant strain. Our results showed that the T3SS mutant strain of ORh26, unlike the wt strain, did not cause a reduction in lesion size and lesions in plants treated with the mutant strain were similar in size to the untreated control group (Figure 4A). This suggests that T3SS is essential for the observed resistance phenotype. Similarly, we found that the number of *P. syringae* pv. *aptata* P21 in the infected leaves was significantly lower in plants treated with the wt strain than in the control group or the group treated with the mutant (Figure 4B). Interestingly, the plants treated with the T3SS mutant strain had a higher bacterial count than the other groups. It is important to note that we did not detect any *P. marginalis* ORh26 in the infected leaves, but only on the sugar beet roots (Appendix A), suggesting that ORh26 did not migrate from roots to leaves and that there was no direct contact between ORh26 and P21 in the infected leaves. This suggests that the observed effects on plant immunity were indirect and mediated by the interaction of ORh26 with the plant roots. To determine whether the observed loss of ISR in the ORh26 Δ*hrcT* mutant was due to a general colonisation defect, we quantified root-associated bacterial populations in plants infected with P21. Colony counts at 7 days post-infection showed no significant difference between the wt and the mutant strain (Appendix A), indicating that the Δ*hrcT* mutation does not affect root colonisation under these conditions. Next, we investigated how deletion of T3SS affected key immune responses such as H_2_O_2_ levels (Figure 4C) and protective enzyme activity (Figure 4D,E) in sugar beet leaves. Deletion of T3SS in ORh26 affected POD activity, as POD was activated to a similar extent in infected plants treated with the T3SS mutant as in the untreated control plants. This was accompanied by higher levels of H_2_O_2_ in these plants, suggesting that the T3SS may be involved in modulating both POD and H_2_O_2_ levels to enhance resistance. In contrast, deletion of T3SS had no significant effect on the induction of PAL activity (Figure 4E), suggesting that PAL is regulated independently of T3SS in ORh26.

### 2.5. T3SS-Dependent and Independent Pathways of Systemic Resistance Gene Regulation in Sugar Beet

To investigate the molecular mechanisms underlying systemic resistance in sugar beet, we examined the expression of four key plant genes and genetic markers of ISR: *NPR1 (NONEXPRESSOR OF PATHOGENESIS-RELATED GENES 1)*, *MYC2 (MYELOCYTOMATOSIS 2)*, *LOX (LIPOXYGENASE)* and *NCED (NINE-CIS-EPOXYCAROTENOID DIOXYGENASE)*, which are central to plant immune and stress signalling pathways [2]. NPR1 is a master regulator of the salicylic acid (SA) signalling pathway, which is crucial for the establishment of systemic resistance. MYC2 is an important transcription factor in the jasmonic acid (JA) signalling pathway, which also plays a role in systemic resistance. LOX encodes lipoxygenase, an enzyme involved in JA biosynthesis that contributes to both local and systemic defence responses. NCED encodes 9-cis-epoxycarotenoid dioxygenase, a rate-limiting enzyme in the abscisic acid (ABA) biosynthetic pathway that induces stomata closure and thus may increase resistance to pathogens that infect the plant via the stomata, such as *P. syringae*. After infection with *P. syringae* pv. *aptata* P21, all four genes were induced, but their expression levels varied depending on the previous treatment with *P. marginalis* ORh26 (Figure 5).

Treatment with the wt strain of ORh26 significantly increased the expression of *NPR1*, *MYC2* and *LOX* in infected plants compared to untreated infected controls. This suggests that the observed reduction in lesion size and bacterial proliferation is related to the activation of SA and JA signalling pathways by ORh26. Interestingly, *NPR1* and *LOX* were also induced in uninfected plants treated with the wt strain, indicating a priming effect that prepares the plant for possible pathogen attack. However, the expression of *NCED* was not significantly altered by ORh26, indicating that the ABA signalling pathway is not activated by this strain.

To determine whether different protective enzyme profiles in beneficial bacteria correlate with different immune responses, we also examined the effects of *P. lurida* ML159 (Appendix A). In contrast to ORh26, ML159 strongly induced PPO activity (Figure 2D), but failed to significantly reduce lesion size in the infected plants (Figure 1A). This distinct enzymatic profile, coupled with its lack of protective effect, made ML159 a useful comparative strain to assess its effect on the induction of ISR-associated marker genes. In terms of gene expression, ML159 did not induce *NPR1* or *MYC2* in the infected plants, consistent with the lack of significant activation of SA-mediated defence mechanisms. However, ML159 significantly increased the expression of *LOX* and *NCED* in infected plants, suggesting that alternative signalling pathways, possibly involving JA-ABA, modulate the immune response, but not to an extent that offers protection against P21.

To assess the role of T3SS in regulating ISR gene expression, we compared the effects of the wt and T3SS mutant strains of ORh26. In uninfected plants, the mutant strain failed to induce *NPR1*, emphasising the requirement of T3SS for the activation of this gene. In infected plants, *NPR1* and *MYC2* were induced, but expression levels were comparable to untreated controls and significantly lower than in plants treated with the wt strain, confirming that T3SS is crucial for their full induction. Interestingly, *LOX* expression was induced in uninfected plants treated with the mutant, but induction was lower in infected plants than in those treated with the wt strain, indicating a partial T3SS dependence for *LOX* activation. This observation supports the hypothesis that the JA and SA signalling pathways are interconnected and influenced by T3SS activity. Finally, deletion of T3SS had no significant effect on *NCED* expression, suggesting that the ABA signalling pathway functions independently of T3SS-mediated regulation in ORh26.

## 3. Discussion

Collections of bacterial isolates from healthy crops are invaluable resources for the discovery of beneficial microbes that improve the growth and resilience of crops under stress conditions. Screening these collections can reveal strains with potential as biostimulants for sustainable agriculture [10], but can also aid the discovery of novel plant–microbe interaction mechanisms. This study is the first to show that T3SS of a plant-beneficial *Pseudomonas* strain can induce ISR in a host plant. While the T3SS is traditionally associated with pathogenicity, our results demonstrate its crucial role in the activation of systemic plant defence by a root-colonising beneficial bacterium, significantly expanding the known functional repertoire of this secretion system. We identified *P. marginalis* ORh26 as a particularly effective isolate for the induction of systemic resistance in sugar beet, significantly reducing susceptibility to *P. syringae* pv. *aptata*, a cause of leaf spot disease in sugar beet and other plant species [18,19]. Other *P. marginalis* strains have previously been associated with the promotion of plant growth and induction of resilience to both abiotic [20] and biotic stress [8]. Interestingly, although several isolates tested were classified as *P. marginalis* (Table 1—ORh26, Ol141, OL170, ML154), only ORh26 was able to induce ISR. This illustrates the functional diversity that can exist between strains of the same species. The observed variation could be due to differences in T3SS effector composition, expression levels or other genetic or regulatory factors not captured by species-level taxonomy. Further comparative genomic analyses between ORh26 and other *P. marginalis* isolates could help to identify the specific determinants responsible for ISR induction.

Our study establishes a unique link between T3SS-dependent ISR induction and systemic enzymatic and transcriptional changes in sugar beet. Importantly, deletion of the T3SS in ORh26 did not affect its rhizocompetence, as no significant difference in CFU counts was observed between the wt and Δ*hrcT* mutant strains colonising sugar beet roots. This finding aligns with previous work showing that deletion of the T3SS in *P. fluorescens* Q8r1-96 similarly did not impair long-term rhizosphere colonisation of wheat and pea [21]. Thus, the loss of ISR-related activity in the mutant is not attributable to reduced root colonisation, but rather reflects a functional role of the T3SS in modulating host immune responses.

Activation of POD, which is crucial for coping with oxidative stress and synthesising secondary metabolites, was T3SS-dependent, a phenomenon not previously observed in beneficial bacteria. POD is a well-known component of the plant immune response and plays a crucial role in oxidative bursts during pathogen attacks [22]. Increased POD activity was observed in response to *P. syringae* infection in tomato [23]. Beneficial *Pseudomonas* species can also enhance POD activity, as shown by *P. putida*, which increases POD on plant roots [24], and *P. fluorescens*, which induces POD and other defence enzymes to protect chickpea from *Fusarium* wilt [25]. The tested isolate ORh26 also induced PAL activity, but not in a T3SS-dependent manner. PAL contributes to broad-spectrum disease resistance, including SA-dependent defence in peppers [26] and SA biosynthesis in soybean during infection with *P. syringae* [27]. PAL induction by beneficial *Pseudomonas* was demonstrated by foliar spraying and microinjection in chickpea, enhancing resistance to *Sclerotinia sclerotiorum* [28] and activation of PAL promoters in tobacco [29]. These results suggest that activation of POD and PAL by ORh26 is likely responsible for the observed ISR. Furthermore, this work is the first to show that T3SS from a beneficial bacterium modulates systemic defence-related proteins in the plant, including NPR1, a master regulator of salicylic acid signalling, and MYC2, a key player in jasmonic acid signalling. NPR1 is targeted by *P. syringae* effectors such as AvrPtoB to suppress plant defence [30]. Functional NPR1 is also critical for ISR, as shown by *P. fluorescens* CHA0-induced resistance in *Arabidopsis* [31], but previous studies did not show that a beneficial *Pseudomonas* strain can induce *NPR1* expression systemically and that it requires a complete T3SS operon for this effect. Similarly, MYC2 has been implicated in rhizobacteria-induced ISR in *Arabidopsis* [32] and modulation of the root–shoot microbiota axis in suboptimal light [33]. Our results also demonstrate that MYC2 is not only an important component of ISR, but also related to T3SS functionality in root-colonising bacteria. The role of LOX in ISR has also been reported. For example, *P. putida* BTP1 induces LOX isoforms in the ISR of tomato [34]. The induction of *LOX* by ORh26 in this study is consistent with its role in ISR activation, and this effect was partially T3SS dependent. In addition, another isolate, *P. lurida* ML159, triggered a different response by inducing PPO activity. Although ML159 reduced lesion size compared to the control, the difference was not statistically significant, and thus, ISR cannot be conclusively attributed to this strain. While ML159 did induce the production of protective enzymes, it did not effectively protect against P21. In contrast, ORh26 exhibited significant protective effects, likely due to differences in the enzymes induced by each strain. ML159 primarily induced PPO, whereas ORh26 induced more POD. This divergence in enzyme profiles may explain why ORh26 was effective in protecting against P21 while ML159 was not. Induction of PPO has been associated with other *Pseudomonas* species, such as *P. fluorescens* [35]. ML159 also induced NCED expression, which is associated with drought stress tolerance [36] and ABA-mediated defence [2,37]. The potential dual effect of ML159 on the induction of protective enzymes and drought resistance deserves further investigation to explore its multifunctional applications in sugar beet agriculture. Additionally, potential implications of T3SS in this alternate mechanism of protective enzyme induction in ML159 will be explored in future studies, which requires further development of tools for mutagenesis of *Pseudomonas* sp.

The T3SS of ORh26 belongs to the Hrc1 family, which is the most widespread among plant-beneficial *Pseudomonas* species, but is also found in the plant pathogen *P. syringae* [7]. To better understand this system, we compared the structure of the T3SS operon, which encodes the structural components of the secretory apparatus, between ORh26 and two other species from the same Hrc1 family. *P. marginalis* SBW25, a species closely related to ORh26, exhibits plant growth-promoting properties [7]. The T3SS in SBW25 is functional, as demonstrated by its ability to secrete T3Es that elicit a hypersensitive response in *Arabidopsis* [15]. The T3SS operons of ORh26 and SBW25 have a similar composition and gene distribution, but ORh26 has two additional genes, *hrcV* and *hrcD*, which encode components of the export apparatus gate and the outer ring of the basal body, respectively. Interestingly, both strains lack the conventional needle subunit that forms the needle filament, as well as the translocon components required for needle tip assembly and effector import into host cells. They retain only the HrcB subunit, which forms the inner rod of the needle. Despite these differences, the functional T3SS in SBW25 suggests that ORh26 may also possess secretory capabilities. Upstream of the *hrpB* gene, where needle-encoding genes are located in pathogenic *P. syringae*, both SBW25 and ORh26 have hypothetical protein-encoding genes. Since needle and translocon components are the least conserved parts of the T3SS [38], it is plausible that some of these hypothetical genes encode needle subunits, suggesting that these strains may have a complete T3SS structure. However, the exact architecture of the T3SS in these *P. marginalis* strains remains to be further elucidated. The use of electron microscopy techniques could provide clarity on whether the needle component of the T3SS is truly absent in this system.

Despite the differences in T3SS structure compared to *Pseudomonas* pathogens, ORh26 carries several T3E genes distributed throughout its genome. Remarkably, both ORh26 and SBW25 contain a pectate lyase effector gene. Pectate lyase is a plant cell wall-degrading enzyme associated with the soft rot pathogenicity of virulent *P. marginalis* strains [39]. Pectate lyase activity has been associated with another T3E: the HrpW effector of the symbiotic bacterium *Rhizobium etli* [40] and the HrpW1 T3E of *Pseudomonas syringae* pv. tomato DC3000 [41]. However, we also acknowledge that pectate lyases can be secreted via the type II secretion system (T2SS) [42], and this remains a plausible alternative. Since the predicted protein does not share close homology with experimentally validated T3SS or T2SS pectate lyases, its secretion route cannot be confidently inferred and will require experimental validation. Since pectate lyase targets the plant cell wall, it is probably secreted into the apoplast rather than injected into the plant cell, which is in line with the proposed needleless structure of T3SS in ORh26. Its role may be to permeabilise the cell wall to facilitate bacterial interaction with the plant cell, or the penetration of other beneficial microbes to the plant cell membrane. The second effector of ORh26 belongs to the HopJ family of effectors in *P. syringae* and shares 96% sequence identity with a homologue in SBW25 as well as with several *P. syringae* strains. However, the function of this effector remains unknown. The third effector is a hypothetical protein with probable DNA-binding activity and is distinct from the effectors of SBW25 and *P. syringae* strains. Future studies should focus on the biochemical validation of these proposed T3Es and clarification of their final destination in the plant cell (apoplast or the cell cytoplasm).

The function of T3SS in beneficial bacteria is still largely unexplored, as it has mainly been studied in the context of pathogenesis. Our results challenge the conventional view of T3SS as a pure virulence factor and show that beneficial bacteria equipped with T3SS can modulate host immunity in a systemically protective manner. The systemic effects we observed, in which root colonisation by ORh26 increased leaf resistance to *P. syringae*, highlight the dual functionality of T3SS: one that can suppress or enhance immunity depending on the microbial context. In rhizobia, T3SS plays a role in the regulation of nodulation [43], while in non-nodulating symbionts such as beneficial *Pseudomonas*, it appears to have immunomodulatory functions similar to those observed in pathogens [7]. For example, *P. fluorescens* Q8r1-96, a beneficial rhizobacterium, possesses three T3Es—RopAA, RopB and RopM—which are transported into the leaves of *Nicotiana tabacum* cv Xanthi where they inhibit ROS production [21]. While this suggests that T3Es may play a local immunomodulatory role in beneficial bacteria, the systemic effects of T3SS in these organisms are still completely unknown. In sugar beet, *P. marginalis* ORh26 had systemic effects on plant immunity and gene expression, and these effects required an intact T3SS operon. Deletion of the T3SS in ORh26 completely abolished the ability to induce ISR, as evidenced by the lack of POD activation and altered H_2_O_2_ levels. In a previous study, we obtained a similar result with another *P. marginalis* isolate, OL141, in which the deletion of T3SS had an effect on the ISR against a less virulent strain of *P. syringae* pv. *aptata*, P16 [8], but in case of ORh26, we could also demonstrate systemic changes in defence gene and enzyme levels, which were linked to T3SS. Compared to our previous work on OL141 [8], which induced ISR only against the less virulent *P. syringae* strain P16, the current study identifies ORh26 as a more potent ISR-inducing strain that confers protection against the agriculturally relevant P21 strain. Therefore, our data suggest that the T3SS can play a role in systemic resistance, likely by modulating oxidative burst and downstream immune responses. Induction of ISR-related genes was also significantly impaired: the mutant strain did not induce *NPR1* and *MYC2*, while *LOX* induction was reduced compared to the wt strain, which emphasises the systemic role of T3SS in modulating both enzyme activity and gene expression, essential components of ISR. However, the validation of the secretory activity of T3SS in ORh26 remains a focus of our future studies. These experiments will help to elucidate if the T3SS role in ISR depends on its structural genes, on its T3Es, or on both components.

Our data support a preliminary model (Figure 6) in which the T3SS of *P. marginalis* ORh26 stimulates systemic defence by either providing immunomodulatory T3SS effectors or acting as a MAMP itself.

Given the lack of annotated needle components, we speculate that structural variations allow the release of effectors directly into the apoplast instead of injection. This model raises several important questions. The first concerns the mechanism of T3SS secretion and the site of MAMP recognition. Since we could not annotate a needle subunit gene, it is possible that the T3SS is truncated in *P. marginalis* ORh26, resulting in the release of effector proteins into the apoplast, as was observed in *P. fluorescens* 2P24 [44]. In this scenario, either T3SS structural components or the effector proteins themselves could act as MAMPs that are recognised by plant surface receptors, such as PRRs. Another possibility is that effector activities, such as pectate lyase activity, generate degradation products that act as DAMPs (damage-associated molecular patterns) and subsequently trigger immune signalling [45]. On the other hand, if the T3SS in ORh26 has a needle filament encoded by yet-to-be-identified genes, the effectors could be injected directly into plant cells and recognised by intracellular receptors, similar to the recognition mechanisms observed in pathogenic interactions [46]. These hypotheses emphasise the need for further studies to clarify the structural and functional aspects of T3SS in *P. marginalis*. While our data indicate a T3SS-dependent ISR response, the secretory activity of the system in ORh26 remains to be demonstrated. Future work will aim to test whether ORh26 actively delivers effectors into plant cells.

Following receptor activation, an immune signalling cascade likely generates systemic signalling. The observed systemic signalling effect could also be a consequence of T3E activity directly within the plant cells, regardless of MAMP/DAMP recognition. For example, T3Es could interact with the cellular machinery of the host by binding to DNA and modulating the expression of genes involved in hormone biosynthesis. The involvement of *NPR1*, *MYC2* and *LOX* in the T3SS-dependent ISR suggests that both SA and JA signalling pathways may be involved in this process, consistent with other ISR mechanisms [2]. Systemic signalling appears to prime leaves for enhanced immunity, as evidenced by sustained *NPR1* induction and subsequent activation of POD following infection. Notably, *NPR1* induction may counteract the immunosuppressive effects of *P. syringae* effectors such as AvrPtoB, which inhibit this protein to evade plant defences [29]. In addition, POD utilises H_2_O_2_ to produce ROS that create an inhospitable environment for pathogens such as *P. syringae* [47]. While our results clearly demonstrate a T3SS-dependent systemic immune response, the exact nature of this response remains to be determined. The limited number of T3SS effectors identified in ORh26 compared to *P. syringae* suggests that classical effector-triggered immunity may not be the primary mechanism. One of the predicted effectors, a pectate lyase, could contribute to apoplastic DAMP signalling and thereby trigger PTI-like responses. Thus, the observed ISR could result from the integration of PTI, DAMP signalling and systemic priming mechanisms. Remarkably, we did not observe any negative effects on plant growth or fitness under our experimental conditions. However, further studies are needed to evaluate possible trade-offs and to assess the long-term robustness of the induced resistance. Although transcriptome-wide profiling methods such as RNA-seq can provide a comprehensive overview of host gene expression, the aim of this study was to determine whether the T3SS of a plant-beneficial *Pseudomonas* isolate plays a functional role in ISR. To answer this question directly, we focused on well-characterised immune marker genes (*NPR1*, *MYC2*, *LOX*, *NCED*) that represent key signalling pathways involved in systemic resistance. Future work will extend this analysis using untargeted transcriptomics and proteomics to uncover downstream signalling networks and effector targets. Understanding the full extent of T3SS function in *P. marginalis* ORh26 is critical to advance our knowledge of plant–bacteria interactions. Elucidating the local and systemic immune signalling mechanisms will not only deepen our fundamental understanding of ISR, but also pave the way for agricultural applications. The utilisation of T3SS-mediated ISR could offer new strategies for sustainable crop protection, improving both disease resistance and plant resilience in an environmentally friendly way.

## 4. Materials and Methods

### 4.1. Bacterial Strains and Media

The T3SS-positive *Pseudomonas* strains used in this study were obtained from sugar beet samples as described in previous studies [8,10] and are listed in Table 1. The strains were cultured in different media, including Luria Bertani (LB), Luria agar (LA) or *hrp*-inductive minimal medium (HIM) [48] at a temperature of 30 °C. LA and LB media were used to propagate the strains, and HIM medium was used to prepare the bacterial cultures prior to sugar beet treatment, as T3SS is expressed in this medium [48]. *P. syringae* pv. *aptata* P21, previously isolated from sugar beet, was used for the infection tests [19]. This strain was cultivated on King’s B medium [49] or LB at 30 °C.

### 4.2. Bacterial Treatment and Infection Susceptibility Assay

Sugar beet (Heston genotype, Maribo) seeds were sterilised with sodium hypochlorite and ethanol, following previously established protocols [10]. Seeds were sown individually in plastic pots (dimensions: 10 × 10 × 10 mm), each containing sterilised soil (Plagron LightMix substrate). The pots were watered regularly and placed in a growth chamber with a temperature of 25 °C and a photoperiod of 16 h of light followed by 8 h of darkness. After two months of growth, plants were treated with bacterial cultures, with each experimental group receiving a single strain listed in Table 1. Each strain was tested in 3 biological replicates, and each replicate contained 8 plants, in order to produce a sufficient amount of plant material per replicate for downstream analysis. Bacterial suspensions were prepared by growing an overnight culture in HIM medium, followed by washing and resuspension in sterile distilled water. A total of 10 mL of this suspension was set to 10^9^ CFU/mL and was added weekly to the soil. The control group was not exposed to bacteria, but the pots were watered with an additional 10 mL of sterile distilled water. The treatment was repeated 3 times over a 2-week period. On the day of the final treatment, plants were infected with *P. syringae* pv. *aptata* P21. P21 suspension prepared in sterile distilled water was adjusted to a concentration of 10^4^ CFU/mL. Subsequently, 100 µL of the suspension was injected into the middle of a sugar beet leaf using a sterile syringe. One first true leaf was infected per plant. Negative control groups received water without bacteria. One week after infection, leaves were harvested and scanned with a computer scanner. Necrotic lesions were quantified using ImageJ software (Fiji bundle, 2.9.0) [50]. *P. syringae* pv. *aptata* P21 was quantified in infected leaf tissue by qPCR as previously described [51] using the Applied Biosystems™ StepOne™ Real-Time PCR System with Maxima SYBR Green/ROX qPCR Master Mix (Thermo Scientific™, USA). The primers used for the quantification of bacteria are listed in Appendix A.

### 4.3. Root Colonisation Assay

To assess bacterial colonisation, sugar beet roots were harvested 7 days post-infection, coinciding with the time of leaf lesion assessment. Roots were first gently washed in sterile distilled water to remove loosely attached soil particles, followed by a vigorous rinse to dislodge surface-associated microbes. Root tissues were then processed for colony-forming unit (CFU) quantification using a previously established protocol, and CFUs were expressed per gram of fresh root weight [52].

### 4.4. Quantification of Hydrogen Peroxide in Sugar Beet Leaves

The content of hydrogen peroxide (H_2_O_2_) was quantified as previously described [53]. In brief, 150 mg of sugar beet leaf tissue was homogenised in liquid nitrogen and 1 mL of extraction buffer (consisting of 0.25 mL 0.1% *w*/*v* trichloroacetic acid, 0.5 mL 1 M potassium iodide and 0.25 mL 10 mM potassium phosphate buffer, pH 5.8) was added. The mixture was incubated for 10 min on ice under light protection conditions. The homogenate was centrifuged at 13,000 rpm for 10 min at 4 °C. The supernatant was transferred to new microtubes and centrifuged again under the same conditions. The resulting supernatant was incubated in the dark for 20 min and the absorbance was measured at 350 nm using a spectrophotometer. The concentration of H_2_O_2_ was calculated using the molar extinction coefficient (0.28 μM^−1^ cm^−1^) and expressed as nmol H_2_O_2_ per gram of leaf tissue.

### 4.5. Preparation of Enzyme Extracts

Leaf samples ground in liquid nitrogen were stored at −80 °C until further use. The frozen samples were kept on ice during processing to prevent thawing. For extraction, 1 g of plant tissue was transferred to microtubes and 1.5 mL of cold Tris-HCl buffer, pH 7.5, was added to ensure complete resuspension. Homogenisation of the tissue was performed on ice and under constant vortexing. The homogenised tissue was centrifuged at 14,000 rpm for 20 min at 4 °C. The supernatant was collected in new microtubes, aliquoted to avoid repeated freeze–thaw cycles and stored at −80 °C. The protein concentration in the enzyme extract was determined using the Bradford method.

### 4.6. Peroxidase (POD) Activity

Peroxidase activity was measured spectrophotometrically using guaiacol as a substrate [54]. The reaction mixture contained 0.5 mL of protein extract, 0.5 mL of 0.1% guaiacol, 1.5 mL of 50 mM Tris-HCl buffer (pH 7.5) and 0.5 mL of 1% H_2_O_2_. A blank sample was prepared without the enzyme extract. The increase in absorbance at 470 nm was observed at 30 s intervals over 3 min. Enzyme activity was expressed as the rate of change of absorbance (ΔA/min) per gram of leaf tissue.

### 4.7. Polyphenol Oxidase (PPO) Activity

PPO activity was determined by monitoring the oxidation of catechol [55]. The reaction mixture consisted of 1.5 mL of 0.1 M citrate-phosphate buffer (pH 6.0), 0.5 mL of proline solution (5 mg/mL), 0.5 mL of catechol (2 mg/mL) and 0.5 mL of enzyme extract. Catechol was added last as an enzyme substrate. The change in absorbance at 495 nm was recorded for 3 min at 30 s intervals. PPO activity was calculated as absorbance change per minute (ΔA/min) and expressed per gram of leaf tissue.

### 4.8. Phenylalanine Ammonia-Lyase (PAL) Activity

PAL activity was determined by measuring the production of trans-cinnamic acid [56]. The reaction mixture contained 100 µL of the enzyme extract, 0.5 mL of 50 mM Tris-Cl buffer (pH 8.8) and 0.6 mL of 1 mM L-phenylalanine. The mixture was incubated at 40 °C for 60 min and the reaction was terminated by adding 1 mL of 2 N HCl. The absorbance was measured at 290 nm. Enzyme activity was expressed as µg of trans-cinnamic acid per gram of leaf tissue.

### 4.9. Genome Sequencing and Annotation of Pseudomonas marginalis Orh26

Genomic DNA was isolated with the Monarch^®^ Spin gDNA Extraction Kit (United Kingdom). The genome of *P. marginalis* Orh26 was sequenced using Macrogen’s Illumina technology. Library preparation was performed using the Watchmaker DNA PCR Free Kit, and sequencing was performed on the NovaSeqX platform in a paired-end configuration. Raw sequencing data quality was assessed using FastQC v0.12.1 [57] and adapter trimming was performed using Cutadapt v4.5 [58]. De novo genome assembly was performed with SPAdes v3.15.5 [59], followed by assembly polishing with Pilon v1.24 [60]. Genome annotation was performed using the NCBI Prokaryotic Genome Annotation Pipeline [61]. The taxonomy of the strain was determined by comparing housekeeping genes with the NCBI genome database using BLAST (2.16.0+). In addition, taxonomic classification of ORh26 was performed using the Type Strain Genome Server (TYGS), which calculates digital DNA–DNA hybridisation (dDDH) values and constructs genome-based phylogenies based on whole-genome sequence comparisons [62]. Secretion systems were identified using MacSyFinder v2.1.3 with the TXSScan model package [63]. Structural genes associated with T3SS and effector proteins were identified using Effectidor [64]. The functional domains of the identified effector proteins were analysed using SMART [65].

### 4.10. Construction of the T3SS Deletion Mutant

The ΔT3SS mutant of *P. marginalis* Orh26 was generated by one-step allelic exchange using the pKNOCK system [66]. A 600 bp fragment within the *hrcT* gene in the T3SS operon was amplified with primers IA089 and IA090 (Appendix A) and then digested with Promega’s restriction enzymes XbaI and XhoI according to standard protocols. The digested insert and plasmid were purified by gel extraction using the GeneJETTM Gel Extraction Kit (Thermo Scientific™). The insert was then ligated into the pKNOCK-Km vector using T4 DNA ligase from New England Biolabs and propagated in *E. coli* CC118. The resulting construct was then introduced into *P. marginalis* Orh26 by triparental mating with an *E. coli* helper strain (pRK2013), as previously described [67]. The mutant strains were selected on LA supplemented with 100 µg/mL kanamycine and 20 µg/mL Irgasan, and their genetic alteration was confirmed by PCR using IA095-IA096 primers to amplify the region around the insertion site. In addition, the identity of the mutant strains was validated by sequencing analysis.

### 4.11. Quantification of Relative Gene Expression Levels

Total RNA was extracted from 100 mg of sugar beet leaf tissue crushed in liquid nitrogen using the Rneasy Plant Mini Kit (Qiagen, The Netherlands) according to the manufacturer’s protocol. Genomic DNA contamination was removed with Dnase I, Rnase-free (Thermo Scientific™). RNA concentration and purity were determined by measuring the absorbance at 260 nm using an Agilent BioTek Epoch 2 microplate spectrophotometer. A first strand of cDNA was synthesised from 1 µg of total RNA using the RevertAid First Strand cDNA Synthesis Kit (Thermo Scientific^TM^) and following the manufacturer’s instructions. Quantitative PCR (qPCR) was performed to quantify the relative expression of the *NPR1*, *MYC2*, *LOX* and *NCED* genes using gene-specific primers (Table 1). Reactions were performed using the Maxima SYBR Green/ROX qPCR Master Mix (Thermo Scientific™). Relative gene expression was determined by calculating the ΔCt values obtained by subtracting the Ct values of the target genes from those of the 25S rRNA gene. The fold change in gene expression levels was calculated using the ΔΔCt method and the untreated and uninfected control group as a reference.

### 4.12. Data Analysis

All experiments were performed with three biological replicates. Statistical analyses were performed using GraphPad Prism version 9.0.0 for Windows. A non-parametric Kruskal–Wallis test was used to assess differences between groups. Post-hoc pairwise comparisons were performed using Dunn’s multiple comparisons test to control for type I error. A significance level of 0.05 was used for all analyses.

## Figures and Tables

**Figure 1 plants-14-01621-f001:**
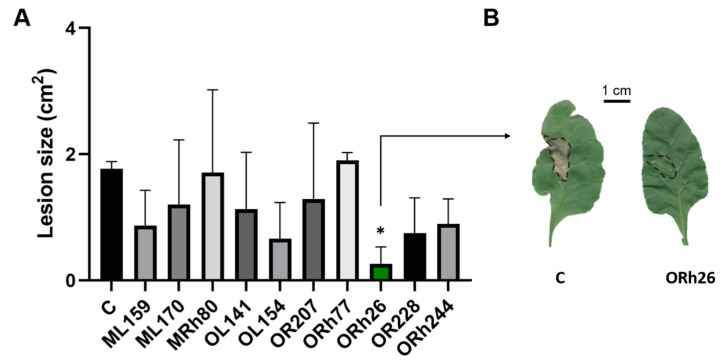
Effect of treatment with different T3SS-positive *Pseudomonas* isolates on the susceptibility of sugar beet to infection with *Pseudomonas syringae* pv. *aptata* P21: (**A**) The species of isolates are listed in Table 1. Sugar beet was treated by applying individual isolates to the soil so that they could interact with the roots. The control group (C) was not exposed to bacteria. After treatment, leaves were infected with *P. syringae* pv. *aptata* P21 and susceptibility was assessed by the size of necrotic lesions at the infection sites. Treatment with *P. marginalis* ORh26 significantly reduced lesion size, indicating induction of systemic resistance (*p* < 0.05 indicated with *). (**B**) Representative necrotic lesions in the control group and in the ORh26-treated group. The lesions in the ORh26-treated group appear as small spots, indicating limited spread of the pathogen, in contrast to the larger, more compact lesions observed in the control group.

**Figure 2 plants-14-01621-f002:**
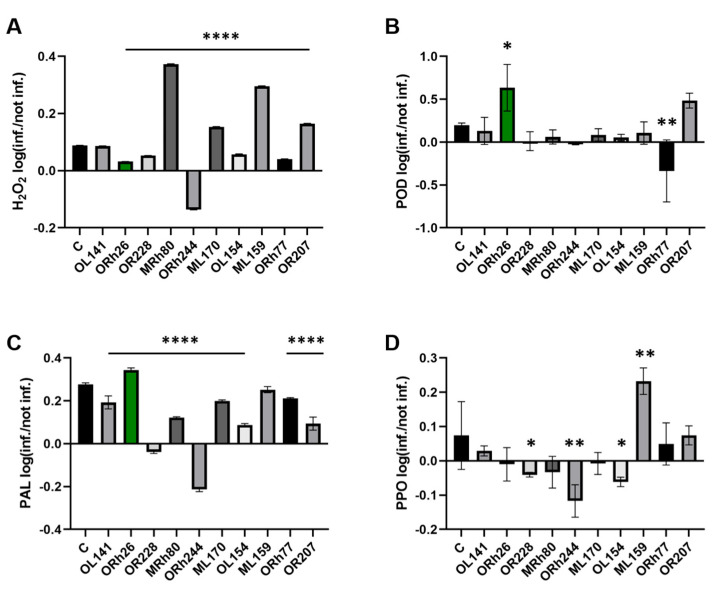
Changes in H_2_O_2_ content and protective enzyme activity in sugar beet leaves after infection with *Pseudomonas syringae* pv. *aptata* P21. Plants treated with different *Pseudomonas* isolates were compared with the control group (C), which was not exposed to bacteria before infection. The leaves were harvested 7 days after infection and the biochemical parameters were analysed in comparison to the uninfected leaves: (**A**) Changes in H_2_O_2_ content; (**B**) Peroxidase (POD) activity; (**C**) Phenylalanine ammonia lyase (PAL) activity; (**D**) Polyphenol oxidase (PPO) activity. Different strains are represented as different coloured bars and ORh26 is highlighted in green. Significant differences of bacterial treatments compared to C are indicated as follows: *p* < 0.05 (*), *p* < 0.01 (**), and *p* < 0.0001 (****).

**Figure 3 plants-14-01621-f003:**
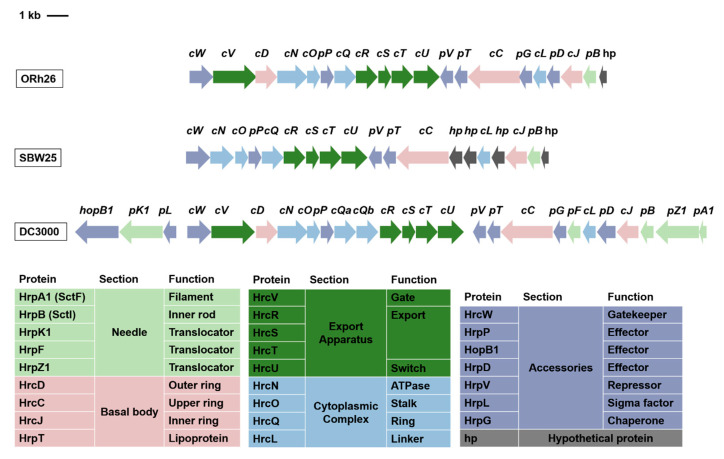
T3SS gene cluster of *Pseudomonas marginalis* ORh26: The genes within the cluster are labelled with the abbreviations *hrc* (*c*) and *hrp* (*p*). The functions of the corresponding proteins are listed in the adjacent table. The operon structure of *P. marginalis* ORh26 is compared with that of *P. marginalis* SBW25 and *P. syringae* pv. tomato DC3000. The colours of the genes are matched with their function listed in the table below.

**Figure 4 plants-14-01621-f004:**
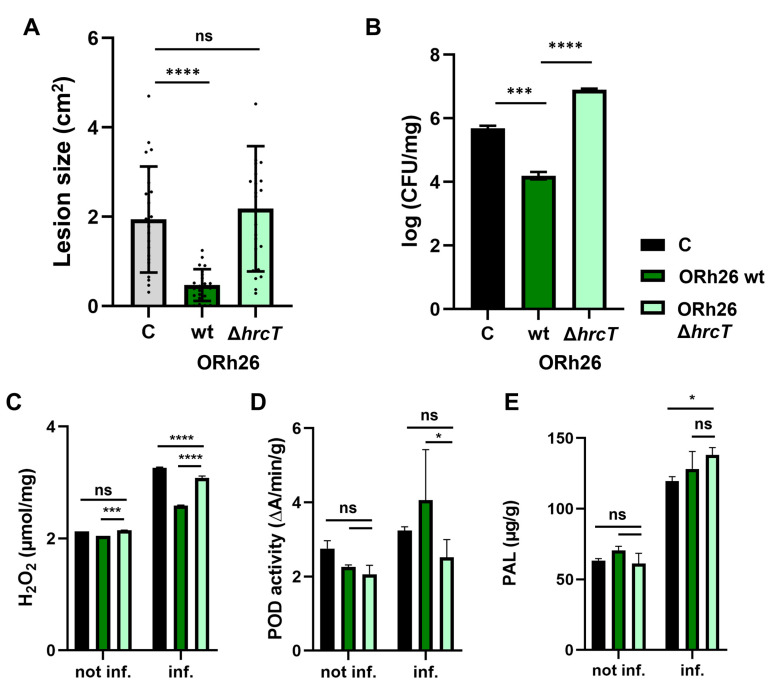
Effect of deletion of T3SS in *P. marginalis* ORh26 on the induction of systemic resistance in sugar beet. T3SS was inactivated by deletion of the *hrcT* gene of the export apparatus: (**A**) Size of lesions due to infection with *P. syringae* pv. *aptata* P21 in sugar beet treated with the wild-type (wt) or the mutant strain (Δ*hrcT*) compared to the untreated control group (C). (**B**) Number of *P. syringae* pv. *aptata* P21 cells in infected leaves. (**C**) H_2_O_2_ levels in infected (inf.) and uninfected (not inf.) plants treated with the wild-type or mutant strain. (**D**) Peroxidase (POD) activity. (**E**) Phenylalanine ammonia-lyase (PAL) activity. Significant differences are indicated as follows: *p* < 0.05 (*), *p* < 0.001 (***), *p* < 0.0001 (****). ns means no significant difference.

**Figure 5 plants-14-01621-f005:**
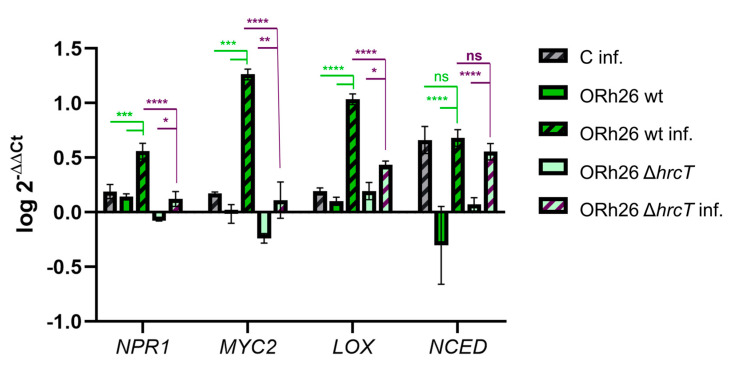
Immune gene expression levels in sugar beet leaves. Plants were treated with *P. marginalis* ORh26 by applying it to the soil to expose the roots, followed by infection of the leaves with *P. syringae* pv. *aptata* P21. Treatments included the wild-type strain (wt), the T3SS deletion mutant strain (Δ*hrcT*) and a control group that was not exposed to bacteria prior to infection (C). Gene expression was analysed in uninfected and infected (inf.) plants. Relative expression levels were normalised to the 25S rRNA housekeeping gene, with the uninfected and untreated control serving as the reference group. Significant differences are indicated as follows: *p* < 0.05 (*), *p* < 0.01 (**), *p* < 0.001 (***), and *p* < 0.0001 (****). ns means no significant difference.

**Figure 6 plants-14-01621-f006:**
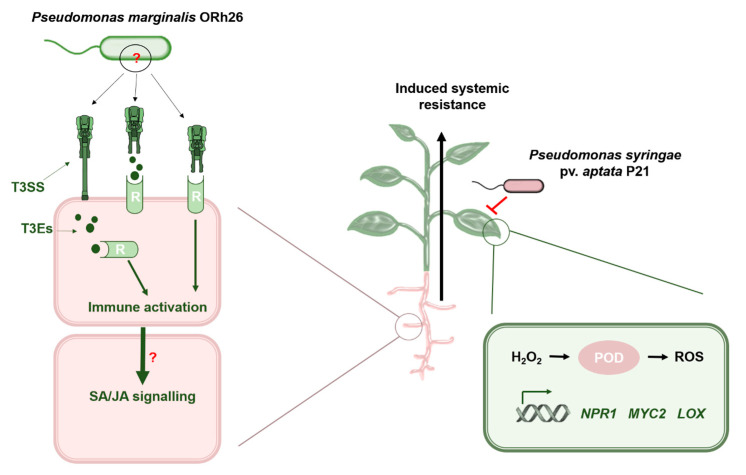
Open questions in the model of T3SS-dependent induction of systemic resistance in sugar beet. In this experimental setup, the plant roots were exposed to the beneficial bacterium *P. marginalis* ORh26, while the leaves were infected with the pathogen *P. syringae* pv. *aptata* P21, so that there was no direct contact between the two. The presence of a functional T3SS in ORh26 reduced the susceptibility of leaves to the pathogen, emphasising its role in systemic resistance. A plausible mechanism is the recognition of T3SS structural components or effector proteins at the root interface. These molecules can be recognised by specific plant receptors (R) and initiate an immune signalling cascade that spreads systemically to the leaves. The final result of this interaction is a systemic effect on immune gene (*NPR1*, *MYC2*, *LOX*) induction and peroxidase (POD) activation which blocks infection. The nature and localisation of T3SS-aasociated MAMPS and the mechanism of signal propagation remain to be identified and are labelled with a question mark.

**Table 1 plants-14-01621-t001:** Names of strains, identification of species and sources of sugar beet samples for the isolated strains. Sample sources include the phyllosphere or rhizosphere of sugar beet, with the month of isolation indicated.

Strain	Species	Sample Source and Month
ML159	*P. lurida*	Phyllosphere in May
ML170	*P. marginalis*	
MRh80	*P. corrugata*	Rhizosphere in May
OL141	*P. marginalis*	Phyllosphere in October
OL154	*P. marginalis*	
OR207	*P. brassicacearum*	Rhizosphere in October
ORh77	*P. arenae*	
ORh26	*P. marginalis*	
OR228	*P. brassicacearum*	
ORh244	*P. kilonensis*	

**Table 2 plants-14-01621-t002:** Putative T3Es in *Pseudomonas marginalis* ORh26.

Locus Tag	Family	Effector Probability Score	SMART/PFAM Domain Accession
ACKCUY_RS25290	Polysaccharide lyase family 1	1	SM000656 SM000710
ACKCUY_RS09560	HopJ type III effector	1	PF08888
ACKCUY_RS17770	Hypothetical protein	0.99	SM000354 PF16422

## Data Availability

*P. marginalis* ORh26 genome is available at NCBI under the accession number JBJPID010000001-JBJPID010000027. All data generated or analysed during this study are included in this published article and its Appendix A. The raw datasets used in the study are available from the corresponding author upon request.

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
