# Peer review of "Type III Secretion System-Mediated Induction of Systemic Resistance by Pseudomonas marginalis ORh26 Enhances Sugar Beet Defence Against Pseudomonas syringae pv. aptata"

_plants, 2025, doi:10.3390/plants14111621_

Round 1
Reviewer 1 Report
Comments and Suggestions for Authors
The entitled " Type III secretion system-mediated induction of systemic re- 2 sistance by Pseudomonas marginalis ORh26 enhances sugar 3 beet defence against Pseudomonas syringae pv. aptata" manuscript focused on the biocontrol Peseudomonas strains and focused on the T3SS genes for the induction of system resistance.
The main concerns:
(1) You performed the genome sequence, why you study the relationship of the biocontrol strain using the single-core genome?
(2) There are lots of strians which belongs to the same speices. There are some different or similarity for the same species, especially the induce systemic resistance.
(3) The author need to perform the RNA-seq to find more different expressed genes to focuse the pathways to answer the questions.
(4) How about the colonization of the mutant compared to the wild-type strain? If the colonization of the mutation strain decreased, it don't act as a biocontrol strain.
(5) The author need to present the general information and the comparative genomic results in the manuscript. The difference between the genome sequences, not only display the T3SS genes in the manuscript.
There are lots of small errors in the manuscript.
(1) For the name of the Pseudomonas, it need italic. Please author check it carefully and revise it.
(2) The name of the biocontrol strain in the manuscript and in the figure should same. Please revise it.
(3) The primers table should put in the Additional file.
(4) The remaining questions, the author need to carefully revise it.
Author Response
Comment 1: You performed the genome sequence, why you study the relationship of the biocontrol strain using the single-core genome?
Response 1: We thank the reviewer for this comment. In the revised manuscript (lines 194-202, 664-668, Supplementary Table 2), we have now expanded the strain classification analysis by incorporating a whole-genome-based taxonomic comparison using the Type Strain Genome Server (TYGS). This approach provides a more robust phylogenetic placement than single-gene analyses. The TYGS analysis confirmed that Pseudomonas marginalis ORh26 belongs to the P. marginalis species, with digital DNA–DNA hybridization (dDDH) values of 74.9% (d0), 69.5% (d4), and 76.5% (d6) when compared to the type strain P. marginalis DSM 13124, along with a G+C content difference of only 0.29%. These results are consistent with species-level identity. We have also added this information as a new Supplementary Table S2 in the revised manuscript.
Comment 2: There are lots of strians which belongs to the same speices. There are some different or similarity for the same species, especially the induce systemic resistance.
Response 2: We thank the reviewer for this important remark. As suggested, we addressed the variability of ISR-related phenotypes between strains of the same species. Although several of the isolates tested were classified as P. marginalis, only ORh26 consistently induced systemic resistance in sugar beet. In the revised manuscript (lines 354–361), we now discuss this intraspecific variability and propose that differences in ISR induction may result from variations in effector protein levels, gene regulation, or other genetic factors beyond species-level classification. This underlines the need for more detailed comparative analyses between functionally divergent strains within the same taxonomic group, which will be the goal of future studies.
Comment 3: The author need to perform the RNA-seq to find more different expressed genes to focuse the pathways to answer the questions.
Response 3: We appreciate the reviewer’s suggestion and fully agree that RNA-seq is a powerful tool for transcriptome-wide analyses. However, our aim in this study was to test the functional involvement of the T3SS in ISR induction by a microbiome-derived P. marginalis isolate. To this end, we followed a targeted and hypothesis-driven approach by monitoring the expression of key ISR-related genes (NPR1, MYC2, LOX, NCED), which represent the major signaling pathways involved in systemic resistance. As mentioned in the revised manuscript, these genes are frequently used as markers for SA-, JA- and ABA-mediated immune signaling and allowed us to detect a clear T3SS-dependent activation of specific defense pathways. RNA-seq could indeed identify additional differentially expressed genes, but alone would not clarify the exact mechanism of interaction, such as localization of effectors or identification of host targets, which would still require complementary methods that are beyond the scope of this study. In addition, all RNA-seq results would require qPCR validation, which we have already performed here for key genes. We also believe that a deeper mechanistic insight is essential and are currently planning a follow-up study using transcriptomics and proteomics to characterize this plant–microbe interaction in more detail. We have clarified these points in the revised discussion (lines 548–561) to explain the scope and limitations of our current approach and outline our future direction.
Comment 4: How about the colonization of the mutant compared to the wild-type strain? If the colonization of the mutation strain decreased, it don't act as a biocontrol strain.
Response 4: We thank the reviewer for this insightful comment. We agree that the differences in colonization efficiency between the wt and mutant strains are crucial for the interpretation of the sugar beet phenotype. Although we had previously measured the number of CFUs on sugar beet roots, we initially decided not to include the data as no significant differences were found. However, in light of the reviewer’s comment, we have now included these results in Supplementary Figure 2, as they are indeed important for assessing the functional impact of T3SS deletion. The methodology used to quantify bacterial colonization has been included in the Materials and methods section (lines 605–611, section 4.3). The corresponding results are now presented in the Results section (lines 258–263, section 2.4) and discussed in the Discussion (lines 363–370). In brief, we found that deletion of the T3SS gene hrcC does not significantly affect rhizocompetence as measured by the number of CFUs. This supports the conclusion that the loss of ISR activity in the mutant is due to a specific functional role of the T3SS and not to an indirect effect caused by impaired colonization. We also refer to previous results showing that deletion of the T3SS in a microbiome strain of Pseudomonas did not alter rhizosphere colonization (Mavrodi et al., 2011), which further supports our interpretation.
Comment 5: The author need to present the general information and the comparative genomic results in the manuscript. The difference between the genome sequences, not only display the T3SS genes in the manuscript.
Response 5: We appreciate the reviewer’s suggestion to provide a broader genomic context for P. marginalis ORh26. In response, we have now included general genomic characteristics of the strain (genome size, GC content, number of predicted coding sequences, comparison to P. marginalis reference strain) in the Results section (lines 195–206).
Comment 6: For the name of the Pseudomonas, it need italic. Please author check it carefully and revise it.
Response 6: Thank you for noticing this. This was corrected in the entire manuscript.
Comment 7: The name of the biocontrol strain in the manuscript and in the figure should same. Please revise it.
Response 7: Strain name was included to Figures 4 and 5.
Comment 8: The primers table should put in the Additional file.
Response 8: The primers table was moved to the supplementary files (Supplementary Table 6).
Comment 9: The remaining questions, the author need to carefully revise it.
Response 9: We thank the reviewer for the comment. As part of the revision, we have carefully re-examined the manuscript and addressed all outstanding issues raised in the reviewer comments. We also reviewed the Introduction and Discussion sections to ensure that all open questions are clearly stated and appropriately contextualized (lines 96-98, 354-361, 363-370, 400-407, 444-449, 481-484, 524-526, 539-561).
Reviewer 2 Report
Comments and Suggestions for Authors
The authors report that lesion size induced by treatment with the pathogenic bacterium Pseudomonas syringae pv. apata P21 was reduced when sugar beet plants were pre-treated with non-pathogenic Pseudomonas strains including P. marginalis Orh26. Plant immune responses induced by P. syringae P21 were altered in plants pre-treated with Orh26, e.g. Peroxidase activity and Phenylalanine ammonium lyase activity were increased in Orh26 pretreated plants compared to controls. The authors next showed that Orh26 contained multiple components of the bacterial type3 secretion system (T3SS) and produced a ΔhrcT mutant that lacked a functional T3SS. Pretreatment of sugar beet with the ΔhrcT isolate did not reduce P. syringae P21 induced lesion size and P. syringae P21 growth, while Orh26 WT did. After pretreatment with the ΔhrcT isolate Peroxidase activity was reverted to control levels while no difference in PAL activity was observed between WT Orh26 and the ΔhrcT isolate. Immune signaling marker genes e.g. NPR1, MYC2 and LOX were induced by Orh26 WT but not ΔhrcT.
I have 2 main concerns:
1) The authors published very similar results found with a different P. marginalis strain (OL141) they found associated with sugar beet. See citation number 8. What makes this study novel compared to previous work?
2) The others treated sugar beet plants with P. marginalis strains weekly. All immune signaling components studied here are induced during PTI and ETI. Generally, PTI induction is weaker and more transient than ETI. The authors only found 3 possible effectors in Orh26, much fewer than in P. syringae. This included a pectate lyase that they propose might be delivered to the apoplast by the partial T3SS of Orh26 and might lead to the production of DAMPs, which in turn induce PTI. It is unclear how robust this induction of immune responses is and what other effects on plant fitness might be observed.
Other comments:
3) Figure 1, please explain why lesion size was used as a proxy for pathogen growth instead of colony forming units (cfu), which shows bacterial growth vs. phenotypic appearance of the disease symptoms.
4) Please review the manuscript to make sure species names are in italics and plant genes are in italics and capitalized.
Author Response
Comment 1: The authors published very similar results found with a different P. marginalis strain (OL141) they found associated with sugar beet. See citation number 8. What makes this study novel compared to previous work?
Response 1: We thank the reviewer for pointing this out. While our earlier study (Nedeljković et al., 2024) identified P. marginalis OL141 as an ISR-inducing strain, it only conferred protection against a weakly virulent strain of P. syringae (P16), and failed to protect against the highly virulent P21 strain, which poses greater relevance for agriculture. The current study was therefore designed to identify a microbiome isolate capable of protecting against P21. In doing so, we identified P. marginalis ORh26 and confirmed its efficacy. Furthermore, this study expands on our previous work by providing a deeper mechanistic analysis of T3SS-dependent ISR, including defense enzyme activity and gene expression profiling, thus offering new insights into the functional basis of beneficial microbe–plant interactions. This was further highlighted in lines 481-484 of the discussion section.
Comment 2: The others treated sugar beet plants with P. marginalis strains weekly. All immune signaling components studied here are induced during PTI and ETI. Generally, PTI induction is weaker and more transient than ETI. The authors only found 3 possible effectors in Orh26, much fewer than in P. syringae. This included a pectate lyase that they propose might be delivered to the apoplast by the partial T3SS of Orh26 and might lead to the production of DAMPs, which in turn induce PTI. It is unclear how robust this induction of immune responses is and what other effects on plant fitness might be observed.
Response 2: We thank the reviewer for raising this important point. We agree that the distinction between PTI, ETI, and ISR, especially when triggered by partial T3SS systems or low numbers of effectors, can be complex. While we identified only a few predicted effectors in ORh26, our functional assays (e.g., defense gene expression and enzyme activities) suggest a T3SS-dependent induction of systemic resistance. Whether this response is best classified as ISR, DAMP-triggered PTI, or a hybrid mechanism remains an open question. We have added a paragraph in the Discussion (lines 539–547) acknowledging this uncertainty and suggesting that future studies, including effector delivery and localization assays, will be necessary to resolve the exact mechanism. We also note that under our experimental conditions, we did not observe negative effects on plant growth or health, though more detailed phenotyping would be needed to evaluate potential trade-offs.
Comment 3: Figure 1, please explain why lesion size was used as a proxy for pathogen growth instead of colony forming units (cfu), which shows bacterial growth vs. phenotypic appearance of the disease symptoms.
Response 3: Lesion size is an appropriate metric for ISR screening, as it directly reflects plant health and immune outcomes, whereas CFU measurements quantify bacterial presence without necessarily capturing the extent of disease symptoms or tissue damage, which are key readouts of induced systemic resistance. Lesion size was also used by other studies to screen for ISR inducing strains (Martel et al., 2020; Ongena et al., 2008; Han et al., 2006). Therefore, in our initial screen comparing multiple bacterial isolates, we used lesion size as a proxy for plant health and pathogen virulence because it provides a rapid and visually quantifiable measure of disease severity. This approach allowed us to efficiently compare treatments under standardized conditions. In the case of ORh26, the focus of our study, we also quantified P. syringae P21 bacterial populations to complement the lesion scoring and better understand the underlying mechanisms of protection. We have clarified this rationale in the revised manuscript (lines 120–123).
Comment 4: Please review the manuscript to make sure species names are in italics and plant genes are in italics and capitalized.
Response 4: Thank you for pointing out this mistake, it has been corrected in throughout the manuscript.
Reviewer 3 Report
Comments and Suggestions for Authors
An understanding systemic effect of root colonization by commensals and how this alters pathogen responses in distant organs is critical for the development of safe plant protection measures. Authors start with a very interesting observation that pretreatment of roots with non-pathogenic Pseudomonas strain ORh26 induces IRS, creating a protective effect against pathogenic Pseudomonas in leaves. This is correlated with an increase in peroxidase activity (POD) and phenylalanine ammonia-lyase (PAL) and a slight decrease in polyphenol oxidase (PPO) and ROS compared to controls. Mutants in the T3SS were not longer able to produce this protective effect and the effects on POD and H2O2 were also dependent on the presence of the T3SS. PAL was induced independently.
ISR was also evaluated using typical marker genes NPR1, MYC2, LOX, NCED. WT ORh26 treatment of roots leads to a higher accumulation of all genes upon infection. Whereas colonization with the ORh26 T3SS mutant shows less induction, but this is non-significant in the case of NCED.
The authors have presented a very interesting set of observations, but did not go deep enough to provide mechanistic insight. Here are a few of the issue I have with the manuscript.
In general, strains are added to the soil, but are the equally able to colonize the root? Can the authors comment or address this experimentally?
Related to this - Line 238 (data not shown) – why is this data not shown? If it is mentioned it should be shown, at least in the supplement
Line 368 – IRS mechanisms of ML159 will be explored in future studies- but ML159 did not significantly affect lesion size – so is this still considered IRS?
Please clarify why MLA159 was chosen for gene expression analysis?
The discussion was carefully worded but very speculative.
I would at least like to see the assay with ORh26 showing the T3SS is functional. If it is not functional, then one would have a strong suspicion that a structural component is recognized.
Since functionality of the T3SS has not been shown I do think the predicted of effectors is approporiate. Why do authors think the pectate lyase is secreted by the T3SS when CWDEs are often secreted by the T2SS?
I think it is important report the presence/absence of other secretion systems possessed by ORh26 as they may account for the residual ISR when plants were treated with the T3SS mutant.
Author Response
Comment 1: In general, strains are added to the soil, but are the equally able to colonize the root? Can the authors comment or address this experimentally?
Response 1: We thank the reviewer for raising this important point. We agree that assessing root colonization, especially to compare it between wt ORh26 and the T3SS deletion strain, is crucial for interpreting our results regarding the induction of ISR in sugar beet. In response to the reviewer’s suggestion, we have now included these results in Supplementary Figure 2. The methodology used to quantify bacterial colonization has been included in the Materials and methods section (lines 605–611, section 4.3). The corresponding results are now presented in the Results section (lines 258–263, section 2.4) and discussed in the Discussion (lines 363–370). Briefly, our findings show that ORh26 colonizes sugar beet roots. Moreover, the deletion of the T3SS gene hrcC does not significantly impact rhizocompetence, as indicated by the number of CFUs. This suggests that the loss of ISR activity in the mutant is due to the specific role of the T3SS, rather than being an indirect consequence of impaired colonization. We also refer to previous studies demonstrating that T3SS deletion in a Pseudomonas microbiome strain did not affect rhizosphere colonization (Mavrodi et al., 2011), which further supports our interpretation.
Comment 2: Related to this - Line 238 (data not shown) – why is this data not shown? If it is mentioned it should be shown, at least in the supplement
Response 2: This data has now been shown as Supplementary Figure 2. We did not include it in the first draft of the manuscript because there was no difference between the wt and the ΔT3SS strain.
Comment 3: Line 368 – IRS mechanisms of ML159 will be explored in future studies- but ML159 did not significantly affect lesion size – so is this still considered IRS?
Response 3: We thank the reviewer for their insightful comment. We agree that the lack of statistically significant reduction in lesion size with ML159 warrants clarification regarding its role in ISR. While ML159 did reduce the average lesion size compared to the control, this difference was not statistically significant (as already stated in line 319). However, ML159 did induce the production of H2O2 and protective enzymes, including peroxidase, phenylalanine ammonia lyase, and polyphenol oxidase (Figure 2). Most importantly, the level of induction for these enzymes was different between ML159 and the ISR inducing strain, ORh26. These results suggest that, although lesion size was not significantly affected, ML159 may still contribute to the induction of immune signaling, but in a way that does not warrant sufficient protection against the tested pathogen. We have reworded this section in the manuscript (lines 400-407, 410, 413) to reflect these findings more accurately.
Comment 4: Please clarify why MLA159 was chosen for gene expression analysis?
Response 4: ML159 was chosen for gene expression analysis because, despite not significantly reducing lesion size, it did induce the production of protective enzymes, specifically PPO. This contrasted with ORh26, which induced POD and PAL and also conferred significant protection against P. syringae P21. By comparing ML159 and ORh26, two strains that differ in both protective efficacy and the immune pathways they activate, we aimed to gain further insight into the molecular basis of strain-specific ISR responses. We have now clarified this rationale more explicitly in the revised manuscript (lines 320-322). This complements the changes made to the Discussion per previous comment (lines 400-407, 410, 413).
Comment 5: The discussion was carefully worded but very speculative.
Response 5: We thank the reviewer for this comment. Our intention in the discussion was to interpret our results in the context of the existing literature while highlighting open mechanistic questions related to T3SS-dependent ISR induction. To our knowledge, this study is the first to link the presence of a T3SS operon in the genome of a microbiome-derived Pseudomonas strain to the induction of systemic resistance in plants. We admit that some aspects of the discussion are speculative, but they are clearly framed as such and are intended to stimulate further investigation in this emerging field. The mechanistic basis for how T3SS contributes to ISR remains unresolved and is beyond the scope of this article, but these questions are important to guide future research. To address this point more clearly and refine our interpretation, we have revised several parts of the Discussion based on these and other reviewer suggestions (see lines 96-98, 354-361, 363-370, 400-407, 444-449, 481-484, 524-526, 539-561). These changes aim to better distinguish between evidence-based interpretations and forward-looking hypotheses.
Comment 6: I would at least like to see the assay with ORh26 showing the T3SS is functional. If it is not functional, then one would have a strong suspicion that a structural component is recognized.
Response 6: We fully agree with the reviewer that demonstrating the secretory activity of the T3SS in ORh26 is an important next step toward understanding its role in ISR. However, as this study focuses on establishing a functional link between the presence of a T3SS operon and ISR induction, detailed mechanistic analyses of secretion activity are beyond the scope of the current work. Demonstrating T3SS secretion in a non-model microbiome isolate is technically challenging and typically requires host-mimicking conditions, which are not trivial to reproduce. In our previous study (Nedeljkovic et al., 2024), we showed that T3SS genes are expressed in contact with sugar beet, and here we demonstrate both root colonization and a clear phenotypic consequence of T3SS deletion on ISR. These findings support the functional relevance of the system. We also note that demonstrating secretion alone would not preclude immune modulation by T3SS structural components, as both MAMPs and effectors may contribute to ISR. We have clarified this point in the revised Discussion (lines 524-526), and our future work will focus on dissecting the T3SS mode of action in planta.
Comment 7: Since functionality of the T3SS has not been shown I do think the predicted of effectors is approporiate. Why do authors think the pectate lyase is secreted by the T3SS when CWDEs are often secreted by the T2SS?
Response 7: We thank the reviewer for this important point. As mentioned above, our prediction that the pectate lyase-like protein could be a T3SS substrate was based on the results of the Effectidor prediction tool, which ranked this protein as a top candidate for T3SS secretion. However, we fully agree that this remains a hypothesis that requires experimental validation. Pectate lyases are frequently secreted via the T2SS. At the same time, several pectate lyase proteins, such as HrpW, have been identified as T3SS substrates whose function is to soften plant cell walls to facilitate needle penetration (Kbitko et al., 2007; Pineau et al., 2021). This is already mentioned in the discussion in line 448. Since our candidate protein does not show significant homology with known T3SS- or T2SS-secreted pectate lyases, we cannot reliably infer the secretory pathway based on sequence similarity alone. Thus, as the reviewer suggests, T2SS is a plausible alternative, and this has now been confirmed in the discussion (lines 444–449). In our future work, we will experimentally investigate both the secretory pathway and the immune function of this protein.
Comment 8: I think it is important report the presence/absence of other secretion systems possessed by ORh26 as they may account for the residual ISR when plants were treated with the T3SS mutant.
Response 8: We thank the reviewer for this thoughtful suggestion. To address this issue, we analysed the ORh26 genome using the MacSyFinder tool to search for components of bacterial secretion systems. We confirm that the T3SS operon we targeted is the only T3SS system in the genome. This supports the specificity of our ΔT3SS mutant and suggests that the observed ISR effects are indeed associated with this system. However, we also identified components of other secretion systems (T1SS, T2SS, T4SS, T5SS and T6SS), consistent with the typical complexity of Gram-negative bacterial genomes. While these systems may contribute to the residual activity of the ISR in the ΔT3SS mutant, effector proteins are generally secreted via a single dedicated pathway, and our mutant is suitable for investigating the functional role of the identified T3SS. These results have been added to the Results and Methods sections (Supplementary Table 4, lines 213-214, lines 667-668).
Round 2
Reviewer 2 Report
Comments and Suggestions for Authors
The revised manuscript addressed my concerns adequately.
Minor comments:
The type 3 secretion system mutant was made by deleting the hrcT gene. The authors mention a ΔhrcC mutation in lines 262 and 365 as well as the figure legend for Supplemental Figure 2.
Author Response
We thank the reviewer for valuable comments that have helped us improve the quality of our manuscript.
Comment 1: The type 3 secretion system mutant was made by deleting the hrcT gene. The authors mention a ΔhrcC mutation in lines 262 and 365 as well as the figure legend for Supplemental Figure 2.
Response 1: Thank you for noticing this typo. It has been corrected in the indicated lines.
Reviewer 3 Report
Comments and Suggestions for Authors
Thanks for addressing my concerns, I am satisfied at the combination of experiments and text changes implemented and for the clarification of some points.
Author Response
We thank the reviewer for very observant and useful comments. It helped us to improve the manuscript.
The reviewer did not raise any other concerns.